# Reliable and Sensitive Nested PCR for the Detection of *Chlamydia* in Sputum

**DOI:** 10.3390/microorganisms9050935

**Published:** 2021-04-27

**Authors:** Martina Smolejová, Iveta Cihová, Pavol Sulo

**Affiliations:** 1Department of Biochemistry, Faculty of Natural Sciences, Comenius University, Ilkovičova 6, 842 15 Bratislava, Slovakia; silonova2@uniba.sk; 2Department of Track and Field, Faculty of Physical Education and Sport, Comenius University, Nábr. arm. gen. L. Svobodu 9, 814 69 Bratislava, Slovakia; iveta.cihova@uniba.sk

**Keywords:** *Chlamydia*, diagnostics, nested PCR, zoonosis, DNA sequence, detection limit

## Abstract

Chlamydia are Gram-negative, intracellular pathogens colonizing epithelial mucosa. They cause primarily atypical pneumonia and have recently been associated with chronic diseases. Diagnostics relies almost exclusively on serological methods; PCR tests are used rarely because in patients with positive ELISA, it is nearly impossible to identify chlamydial DNA. This paradox is associated with DNA degradation in sputum samples, low abundance, and low sensitivity of PCR systems. In a newly designed and validated “nested” PCR (NPCR) assay, it was possible to amplify DNA of Chlamydia known to infect humans in 31% samples. The reliability of the assay was confirmed by DNA sequencing, and all PCR products belonged exclusively to the Chlamydiales, mainly recognized as *Chlamydia pneumoniae*. Three samples were related to Ca. *Rhabdochlamydia porcellionis* and Ca. *Renichlamydia lutjani*, which infect arthropods. In one case, samples were taken from sick individual, indicating the potential as a human pathogen.

## 1. Introduction

*Chlamydia pneumoniae* is believed to be responsible for around 5% to up to more than 40% of lower respiratory tract infections (LRTI), known as atypical pneumonia or community-acquired pneumonia [1,2,3]. The numbers vary with age, geographical location, the population studied, and, especially, the diagnostic methods used [3]. In addition, most *C. pneumoniae* infections are probably mild or asymptomatic [2]. Laboratory methods currently used for the diagnosis of acute *C. pneumoniae* infection include culture, immunohistochemical assays, serology, and PCR; the latter two are the most often applied ones [2,4]. Evidence for the presence of specific antibodies in blood serum is the most commonly used method of laboratory diagnosis of *C. pneumoniae*. The microimmunofluorescence (MIF) test and the enzyme-linked immunosorbent assay (ELISA) are currently considered the “gold standards” [2,3,4]. However, the problem is the high prevalence of antibodies against *C. pneumoniae*, especially in the adult population, which is 50% at age 20 and 70–80% at ages 60–70 years [5,6,7], respectively (50–75%) [4]. 

Here, DNA amplification, specifically real-time PCR, appears to be the most promising technology in the development of a rapid, nonculture method for detection of *C. pneumoniae*. More than 25 in-house PCR assays from clinical specimens such as nasopharyngeal (NPS) or throat swabs, sputa, or pleural fluid have been reported in the literature until 2015 [4,8,9]. None of these assays is standardized or extensively validated [4]. In addition, there are extreme differences between serology and DNA amplification assays [2,3,8]. Despite the good specificity, all published PCR-based techniques to detect *C. pneumoniae* in clinical samples have a much lower sensitivity than serology methods such as MIF or ELISA [10,11,12,13,14,15].

There are several PCR modifications that help to increase the sensitivity and specificity of detection. For example, nested PCR (NPCR) involves two rounds of PCR reactions, with the first round targeting a wide DNA region and the second round targeting a narrower sub-region of the products of the first round, which are used as a template [16,17,18].

Any bacterial species can be easily amplified from the conserved regions of 16S rRNA using universal primers, which is widely used in taxonomy and phylogenetic studies [19,20]. Besides the conserved regions, 16S rRNA contains hypervariable regions that are highly specific for a biological species or genus [21,22,23]. Therefore, the aim of the present study was to find a reliable and sensitive nested PCR method for identifying all Chlamydia known to infect humans (*C. abortus*, *C. pneumoniae*, *C. psittaci*, and *C. trachomatis*) from sputum [4,24].

## 2. Materials and Methods

### 2.1. Patients and Samples

Sputum samples from 10 patients with pneumonia (mostly atypical) were obtained from the HPL s.r.o. Microbiological Laboratory (Bratislava, Slovakia) for routine plating tests (four males and six females; mean age: 42; range, 24 to 68 years). Another 156 samples were provided by volunteers from the Faculty of Natural Sciences and the Faculty of Physical Education and Sports (Comenius University, Bratislava, Slovakia). The study was approved by the Ethics Committee of the Faculty of Physical Education and Sport (8/2019 from 21 May 2019). The participants provided written informed consent in accordance with the Declaration of Helsinki. The samples were collected in 2018 and 2019, transported to the laboratory, and stored in a freezer at −20 °C. The *C. pneumoniae* TWAR-183 suspension was purchased from the Deutsche Sammlung von Mikroorganismen und Zellkulturen (DSMZ), Leibniz Institute, and stored at −70 °C. The DNA from *Chlamydia trachomatis* DSM 19411 CP002054; *Chlamydia psittaci* DSM 27007 CP002807, and *Chlamydia abortus* DSM 27085 NC-004552.2 was kindly provided by Dr. Tomáš Szemes (Comenius University Science Park, Bratislava, Slovakia).

### 2.2. DNA Analysis

The DNA was obtained from sputa according to Freise et al. [25,26], with minor modifications. To 0.5 mL of sputa, 200 μL of 50 mM NaOH was added. The samples were vortexed and incubated for 15 min at 95 °C using an Eppendorf Thermomixer Comfort apparatus under mild shaking. Then, 32 μL of 1 M Tris-HCl pH 7.0 was added. The samples were vortexed and centrifuged for 1 min at 14,000× *g*. The pH was checked and was in the range of 6–7. Isolated DNA was stored at −20 °C.

### 2.3. PCR

The primers used in this work are listed in Table 1. The DNA was amplified by FIREpolDNA polymerase FIREPol^®^ DNA Polymerase (Solis BioDyne, Tartu, Estonia) in a 25 μL solution containing 1× buffer B, 0.2 mM dNTPs, 2.5 mM MgCl2, 1× S solution, 1 pmol/μL of each primer, 0.5 μL of DNA, and 0.04 U/μL of enzyme (461-bp amplicon). In the amplifications of 121 bp amplicon, S solution was omitted. The second PCR reaction included the same content, but the 1 μL of DNA from the first reaction was added. Amplification was performed in a DNA Thermal Cycler (Eppendorf Mastercycler 5330, Eppendorf-Nethel-Hinz GmbH, Hamburg, Germany); the thermal cycles for different primers are listed in Table 2. Real-time PCR was performed using the *Chlamydia pneumoniae* PCR Kit© (GeneProof a.s.) according to instructions of the manufacturer, using the StepOne™ Real-Time PCR System. *C. pneumoniae* presence was indicated by FAM fluorophore fluorescence growth. The target gene was the *ompA* gene, product size was about 120 bp (personal communication), and sensitivity was 0.647 cp/μL (Instructions for Use).

### 2.4. Analysis of PCR Products and Their DNA Sequences

The size of the PCR products was determined by electrophoresis in 2% agarose gels and 1× TBE solution. We used λ/PstI as DNA size marker and for the estimation of DNA concentration in PCR. All PCR products were sequenced after EXOSAP-IT treatment, using the ABI–100—Avant and BigDye R Terminator v 3.1 Cycle Sequencing Kit (Applied Biosystems). Sequences were edited in CHROMAS, trimmed to the same size, and compared with each other or with the sequences in the GenBank database using the BlastN program at the National Center for Biotechnology Information (NCBI). Sequence divergence was determined using the ClustalW program [27], which is part of the CLC Genomics Workbench 9.0 program. Phylogenetic relationships were analyzed by the Maximum likelihood phylogeny PhyML (multiple sequence alignment program) included in the CLC Genomics Workbench 9.5 package.

## 3. Results

### 3.1. Pros and Cons of Published Nested PCR Assays 

To elaborate a reliable and sensitive NPCR method for identifying all Chlamydia known to infect humans, we surveyed databases using the search term “nested PCR and Chlamydia”. We found 20 papers that reported 10 different NPCR systems; some of them are listed in Reference 8. These assays were evaluated from the point of view of specificity and efficiency of the PCR in silico (Table 3). 

The primers were restricted to three genes, mostly to *ompA* and 16S rRNA. Most of them were designed exclusively to *C. pneumoniae* and thus cannot be used for the identification of all known chlamydial human pathogens. In addition, most of the primers were polymorphic in the binding site even in *C. pneumoniae.* Sometimes, the primer Tm difference was larger than the recommended 4 °C. Occasionally, primers were too short, and consequently, their Tm was too low. These differences should affect amplification efficiency and might be the reason why positive samples are not amplified. Additionally, only a few of these amplification reactions were tested on a complex specimen such as sputum, and the authenticity of PCR products was seldomly confirmed by sequencing. Only one set of primers [45,46] could be used to detect all recognized chlamydial human pathogens. Unfortunately, the gene (*ompA*/MOMP) codes for major outer membrane protein, a characteristic antigenic component with an extremely high mutation rate [51] (Appendix A). Consequently, this gene is not a good choice to design primers specific for the four chlamydial species. Therefore, we evaluated pan primers for real-time PCR designed to identify the entire genus [50]. Unfortunately, also in this case, primers were designed in conserved regions of 16S rRNA and, consequently, were nonspecific, which was confirmed experimentally. The sequencing of 10 PCR products amplified from sputum and subsequent comparison with the GenBank database revealed in six samples more than 97% identity to *Schaalia odontolytica* (pathogen isolated from carious lesions of the human dentine) and in two samples more than 97% identity to *Actinomyces* sp.; one sequence was unreadable. Only in one case, DNA specific to Chlamydiales was found.

Several NPCR amplification systems have also been published for *C. trachomatis* [52,53]. Unfortunately, primers have the same cons. They were designed to polymorphic sites in 16S rRNA or *ompA*/MOMP, where especially *ompA* is extremely variable (Appendix A).

Apparently, most of the published NPCR assays are neither suitable nor efficient enough to detect all four Chlamydia species in low-density specimen. Additionally, none of these amplification reactions were tested in a complex specimen such as sputum or saliva, and the authenticity of the PCR products was not confirmed by sequencing.

### 3.2. Primer Design, Optimization 

Our primary objective was to design suitable primers for NPCR, which would universally identify the species of *Chlamydia* infecting humans. As a target area, we chose 16S rRNA for a number of reasons. This is the gene present in each cell and is used for basic taxonomy identification as well as for studying phylogenetic relationships. Thus, in the case of non-specific PCR products, it is easy to determine the interfering biological species based on a different sequence and to design new, better primers. Primers have been designed to anneal to hypervariable regions specific for a particular species or genus [18,20,21], which was already been described for *H. pylori* [23]. 

To identify the appropriate sector, sequences of *C. abortus*, *C. pneumoniae*, *C. psittaci*, and *C. trachomatis* 16S rDNA were compared to their counterparts from *Escherichia coli*, *Shigella* sp., *Yersinia* sp., and *Salmonella* sp. species, known to cause reactive arthritis [52]. Sequence alignment in BlastN or ClustalX identified several *Chlamydia* unique regions to which primers were designed. Rough pre-primers were trimmed to have strong “mismatch” at the 3′ end and to maintain the GC content at the 3′ end below 50%. Primer length was then adjusted to maintain the Tm calculated by the Vector NTI v. 10 (InforMax, Inc.) program at around 55 °C (Appendix A). Finally, primer specificity was examined in silico by BlastN comparison to the other GenBank sequences. Primer sequences were selected only when they matched within the Chlamydiales but contained extensive mismatch at the 3′ end of sequences from the other bacterial species. This can be simply done if the Chlamydiales taxid is excluded from the search. Finally, several pairs of primers were designed to amplify longer, about 450 bp segments, and several pairs of primers to amplify shorter regions of about 120 bp.

As a first step, we examined some primer pairs, of which two pairs with the best performance were selected for the amplification of long NPCR (LNPCR) products (461 bp; PNEU S/PNEU Nok and Chp up/Chp down) and other two for shorter NPCR (SNPCR) products (121 bp; PNEU S/Short down and Short up/Chtinok). Then, we optimized the amplification conditions for both reactions using DNA isolated from spare samples of sputum tested as negative that were spiked with a serial dilution of the *C. pneumoniae* TWAR-183 culture. In a number of experiments, we altered the annealing temperature (50–60, increments of 1 °C), the magnesium concentration (from 1 to 3.5 mM), and the number of cycles (25–45). The highest yield was obtained at an annealing temperature of 50 °C for 35 cycles of the most amplification reactions (Table 2) and 50 °C for 30 cycles for Short up/Chtinok-derived PCR (Table 2). At the end, the efficiency of the amplification was verified on the DNA from rest of the examined Chlamydial species.

### 3.3. Nested PCR Is Highly Sensitive—Detection Limit

A weak point of all identification methods is the absence of a detection limit, which should be understood as the minimal number of cells that are reliably identified. This determination can be accomplished by adding (“spiking”) a known amount of *C. pneumoniae* cells directly to the PCR reaction (“colony PCR”) (cells, Table 4) or the spare samples of sputum from volunteers previously tested as negative (sputum, Table 4). An example of such an experiment when DNA is amplified by NPCR is shown in Figure 1. The NPCR is prone to contamination because the tubes are opened to add the second round of primers and reagents [17]. To avoid this contamination problem, a negative control was introduced after each sample (Figure 1). Only cases with a signal in the sample and no signal in the negative control were considered positive.

For better understanding and reproducibility, sensitivity was calculated in three ways: (i) as a dilution equivalent of the original DSM culture, (ii) as the density of cells in the sample, and (iii) as the number of bacteria in the reaction vial when the DNA is still amplified (Table 4). The detection limit for SNPCR was 10× lower than that for long NPCR. The original *C. pneumoniae* titer declared by the provider, according to fluorescence staining, was 5 × 106 cells/mL, which we confirmed by fluorescence staining [23]. Interestingly, our nested PCR assay was 100 to 1000 times more sensitive than the real-time PCR.

### 3.4. Solving Technical Pitfalls—Scale of DNA Degradation 

The other keystone that might be the source of false negatives is the quality of the sample. Sputum samples are considered of good quality if they have ≥10 leukocytes with mucus per low-power field [54]. *Chlamydia* are intracellular parasites, and therefore, to evaluate the presence of human cells and the rate of DNA degradation, we amplified the 1023 bp region of the control region specific to human mitochondrial DNA (mtDNA) [55]. Only samples with a clear PCR signal were selected for further analysis. In early experiments, we were able to amplify chlamydial-specific DNA for the 461 bp amplicon and confirmed the origin by sequencing. However, in routine diagnostics, it might become necessary to reevaluate previously examined samples. Sometimes, we did not obtain a sufficient amount of PCR product to achieve a good-quality sequence. Therefore, we had to amplify these and other positive samples again from DNA preps stored at −20 °C, but the results were negative. This problem has already been described, and the detection limit decreases 100-fold after 4 months of storage at −20 °C [25]. The most plausible explanation was that the DNA in sputum as well the stored DNA is degraded. Indeed, 1 year of storage of DNA isolated by alkaline lysis reduced the amplification yield of mtDNA in all samples (Appendix A). To bypass this problem, DNA specific for *Chlamydia* was amplified by primers designed for the shorter 121 bp amplicon (Table 1). In this case, the PCR product was observed reproducibly from all previously positive samples. All PCR products were sequenced, and comparison with GenBank database sequences confirmed their chlamydial origin. Therefore, an alternative test for degradation was introduced to amplify the 250 bp amplicon of the mtDNA control region (Table 1) [2,56]. Only samples positive in the later test were subjected to further analyses.

### 3.5. Detection of Chlamydial DNA in Sputum 

To evaluate the potential of the NPCR assay in diagnostics, we characterized overall 166 sputa from three different groups. The first group consisted of 10 patients with pneumonia (mostly atypical), confirmed by a diagnostic laboratory (four males and six females; mean age: 42; range, 24 to 68 yr.). Case subjects were defined as those having a fever (temperature 38 °C or more) and a cough that persisted for 3 or more days, clinically confirmed pneumonia, and positive for C. *pneumoniae* ELISA. Three volunteers had chronic pneumonia without fewer and also displayed positive ELISA tests. The remaining 153 sputa samples were provided by healthy volunteers from the Faculty of Natural Sciences and the Faculty of Physical Education and Sports). In four samples, we were not able to amplify mtDNA, and these samples were excluded from further analysis. Four short NPCR products were amplified, albeit with a low yield; therefore, the origin of the DNA was not confirmed by sequencing. One sample provided unreadable sequence. All were excluded from overall analysis, and further sampling was required. Of 157 samples, only in 14 could we amplify the 451 bp long region by LNPCR, but in 48 samples, smaller 121 bp amplicon was present (Appendix A), which provided 31.2% positivity. The origin of all PCR products was confirmed by sequencing. Eleven sequences of LNPCR did not differ from the known *C. pneumoniae* sequences stored in GenBank by more than 3%, the generally accepted level used for the definition of bacterial species [20,57,58]. The other three cases were related to the *Rhabdochlamydia porcellionis* clade (Figure 2 and Appendix A). All sequences derived from the short amplicon were related to *C. pneumoniae*, but due to the size and conserved nature of the region, it was impossible to identify the related *Chlamydia species*. Only the presence of *C. trachomatis* and *C. psittaci* could be excluded (Appendix A).

## 4. Discussion

The identification of all *Chlamydia* known to infect humans is marked by the same stigma as the diagnostics of *Helicobacter pylori* [23], where it is believed that immunochemistry-based detection is more sensitive than PCR, although PCR is generally considered a more sensitive method.

Analysis in silico of the previously published methods revealed their drawbacks. Most of them were designed exclusively for *C. pneumoniae* detection, with the aim to prove it as a chronic disease agent. The number of primers pairs could not be used due to the extremely high polymorphism in *ompA* gene or because of the low specificity, amplifying also nonchlamydial sequences. Therefore, we elaborated new NPCR assays to target 16S rRNA, where it is easy to determine the interfering biological species according to the DNA sequence, allowing a better primer proposal. Primers were designed to hypervariable species-specific regions [20,21], as described in the case of *H. pylori* [23]. Their specificity was proved by the sequencing of PCR products, where all provided sequences falling to Chlamydiales, and most of them were identical to *C. pneumoniae* varieties. We also examined primers degenerated in the polymorphic regions, but the amplification reactions were less efficient in sputum samples than original ones. Therefore, only original primers were used in further work.

Additionally, both NPCR assays were validated using DNA prepared from a complex specimen such as stool, which contains thousands of different bacterial species [60], and saliva, where more than 500 different species are present [61]. No DNA was amplified with *Chlamydia* specific primers from five stool and saliva samples used for the identification of *H. pylori* [23]. However, when samples were spiked with the diluted *C. pneumoniae* suspension, it was possible to amplify DNA that can exclude inhibition in nonspiked samples.

Validation of PCR tests is mostly analytical, and clinical samples obtained with the addition of a known amount of *Chlamydia* are rarely used [2]. Therefore, we validated our NPCR assays by adding a known amount of *C. pneumoniae* in decimal dilutions to previously negative samples. Sensitivity was calculated in three ways, but most importantly, as a dilution equivalent of the original DSM culture. This parameter allows simple comparison with other assays. Surprisingly, taking into account the isolation procedure and dilutions as low as 0.0034 of cell equivalent in PCR reaction can be reliably identified in SNPCR (Table 4). Apparently, the presence of large amounts of fragmented DNA and debris of lysed elementary bodies in the original bacterial suspension is the most plausible explanation for this paradox. Indeed, when we used DSM culture washed twice in PBS instead, the threshold value was 0.34 of cell equivalent in the PCR vial. This assay is at least 100 times more sensitive than real-time PCR, which was also confirmed for six randomly picked positive samples in SNPCR; all were negative in real-time PCR. Apparently, the occurrence of chlamydial DNA was below the real-time PCR detection limit. The actual occurrence of target molecules can be estimated by amplifying gradual dilutions of DNA preparations. In most of the samples, reproducible SNPCR amplification could not be obtained after 5× dilution, indicating that the threshold occurrence of target molecules is nearly the same as the detection limit (Table 4). 

Increased sensitivity also has is disadvantage, namely the higher risk of contamination and false positives [2]. To avoid this, we included one or two negative controls after each sample containing the DNA template. To eliminate the spray effect, we used oil in the first PCR reaction, and the individual operations were carried out in three different rooms. The first was reserved for DNA isolation, the second for pipetting PCR reactions, and the third for electrophoresis and DNA analysis. In addition, pipettes and centrifuges were devoted to each operation and cleaned and sterilized every 3 months. With this setting, it was possible to reproducibly identify chlamydial DNA in any sample that passed the quality test. 

The other problem associated with DNA degradation is degradation during sample storage. We tried to solve this issue by replacing the alkali lysis isolation procedure [25,26] by another DNA isolation procedure, using the SiMax™ Genomic DNA Extraction Kit (SBS Genetech, Beijing, China), the QIAamp DSP DNA Genomic Kit (Qiagen, Valencia, CA, USA), and the DNA-sorb B Kit (Amplia s.r.o). However, as in the case of synovial fluid, all silica-based methods dramatically reduced the ability to identify *Chlamydia*-specific DNA from our sputum samples [25]. The quality of sputum or bronchial lavage samples is another disputed issue that may have a significant impact on the reliability of the assay. Often due to careless handling, DNA is degraded, and as a consequence, it is impossible to amplify any DNA. This problem was solved by the implementation of the mtDNA amplification assay for 1023 bp or 250 bp amplicons. Samples from which mtDNA could not be amplified at all were excluded from further testing.

Based on a literature search, sputum or an NPS may be the preferred specimen for the detection of *C. pneumoniae* by PCR [2]. The DNA amplification methods are considered less sensitive than serology [3,11,12,13]. Padalko et al. [11] analyzed the yield of PCRs for the detection of *C. pneumoniae* in respiratory specimens. Their data were based on routine analysis of respiratory samples submitted for *C. pneumoniae* detection, collected in four large Belgian hospitals during 2 consecutive years. Only 0.2% of the 3560 samples were *C. pneumoniae*-positive. Similarly, Miyashita et al. [12] and Noguchi et al. [13] were not able to amplify *C. pneumoniae*-specific DNA from seropositive patients. Apparently, the most plausible explanation is the insufficient sensitivity of PCR methods. When our SNPCR was used in all samples with positive or threshold serology, it was possible to identify *Chlamydia*-specific DNA (Appendix A).

Despite all progress in DNA amplification, the weakest link of the elaborated assays is the question whether only four species, namely *C. trachomatis*, *C. pneumoniae*, *C. psittaci*, and *C. abortus*, are capable of infecting humans [24,62]. Apparently, *C. psittaci* and *C. abortus* do not play a significant role in pneumonia, since we did not find their DNA in all positive SNPCR samples (Appendix A). The low mutation/polymorphism rates in these sequences did not allow to distinguish *C. trachomatis* and *C. pneumoniae* from other Chlamydiales. Sequencing of 15 LNPCR products proved their *C. pneumoniae* origin. Surprisingly, sequences from three samples did not belong to either of the four assayed species. In recent years, several new types of *Chlamydia* have been found in human samples from volunteers without disease symptoms [63,64] or patients with respiratory diseases [47,65,66]. Sample 1080 contained DNA related to Ca. *Renichlamydia lutjani*. This is a Gram-negative bacterium from the “Chlamydia-related bacteria” family, which was the first reported case of chlamydial infection in organs other than fish gills [67,68]. Apparently, these bacteria are pathogenic also to humans because a sample was taken from a patient with acute pneumonia. Interestingly, this infection can be recognized by a serology test designed for the detection of *C. pneumoniae* antibodies, apparently due to cross reactivity. The DNA from two other samples, 3014 and 57SP, were related to the species from the Ca. *Rhabdochlamydia porcellionis/Rhabdochlamydia crassificans* clade, which infect arthropods [68,69].

## 5. Conclusions

We elaborated a reliable and sensitive nested PCR method for identifying all *Chlamydia* species known to infect humans (*C. abortus*, *C. pneumoniae*, *C. psittaci*, and *C. trachomatis*) from sputum. The method is as sensitive as serology, preferentially used in clinical diagnostics. Sample quality, DNA degradation, and the low sensitivity of PCR assays are the reasons why in samples from patients with positive ELISA, it was nearly impossible to identify chlamydial DNA. The reliability of the elaborated assay was confirmed by DNA sequencing, and all PCR products belonged exclusively to the Chlamydiales, mainly recognized as *C. pneumoniae*. Three samples were identified as zoonosis because the amplified DNA was related Ca. *Rhabdochlamydia porcellionis* and Ca. *Renichlamydia lutjani*, which infect arthropods. This section is not mandatory but can be added to the manuscript if the discussion is unusually long or complex.

## Figures and Tables

**Figure 1 microorganisms-09-00935-f001:**
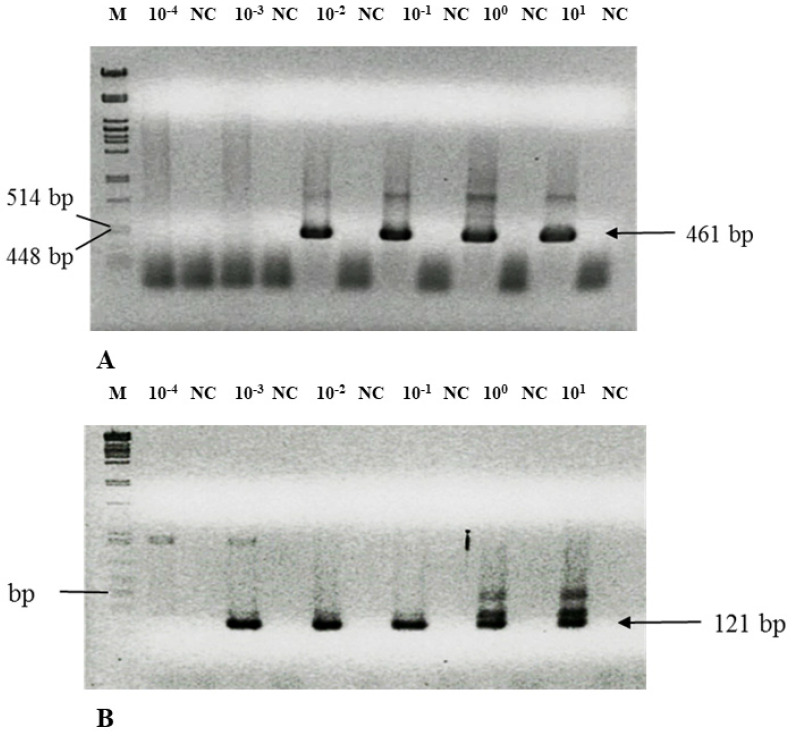
Threshold value of the NPCR assay for *C. pneumoniae* detection in sputa. Lines: M λ/*Pst*I size marker; 500 µL of negative sputum was spiked with 1 µL of 10^−4^–10^1^ dilution of original culture *C. pneumoniae* TWAR-183; DNA was isolated as described in the Materials and Methods section, and 0.5 µL was added to the first PCR reaction, amplified by external primers, and 1 µL was amplified in the second reaction; 10^−4^–10^1^ serial dilutions; NC negative controls. (**A**) PNEU S/PNEU Nok; Chp up/Chp down primers. Size of PCR product is 461 bp. (**B**) PNEU S/Short down; Short up/a Chtinok primers. Size of PCR product is 121 bp.

**Figure 2 microorganisms-09-00935-f002:**
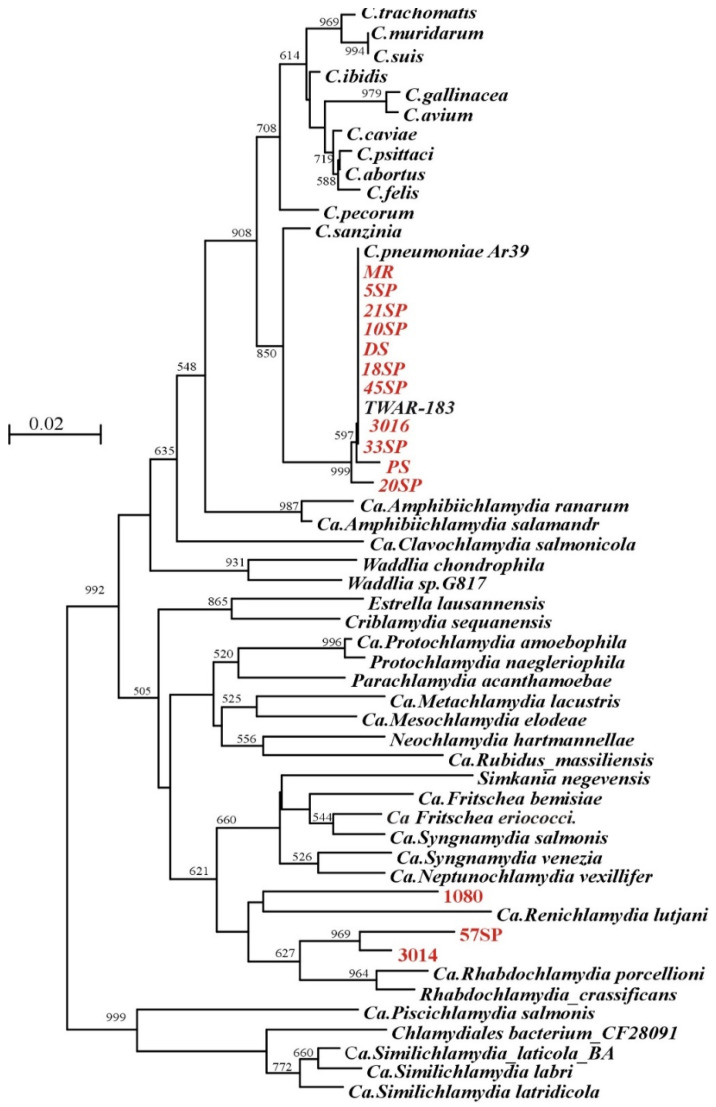
Phylogenetic tree showing the phylogenetic relationships of species of the order Chlamydiales and our specimens (“neighbor-joining algorithm”). A tree based on a comparison of a 346 bp amplified 16S rDNA region flanked by Chp up and Chp down primers. Clinical samples are marked in red. The length of the individual branches corresponds to the extent of the divergence. The stability of the individual branches was evaluated using the “bootstrap” method with 1000 repetitions, and only values greater than 50% are given. Sequences of *Chlamydia* species were derived from [59].

**Table 1 microorganisms-09-00935-t001:** PCR primers.

Primer	Sequence 5′→3′	Target/Size	T_m_ (°C)
Chp up	CATACTTGATGTGGATGGTCTCAACC	16S rDNA 461 bp	54.2
Chp down	GATTTGCTCCATCTCACGATCTT		54.0
PNEU S	TGGGGAAAAGGGAATTCCAC	16S rDNA 635 bp	55.1
PNEU Nok	GGGGCTAGCTTTTAGGATTTGC		55.3
PNEU S	TGGGGAAAAGGGAATTCCAC	16S rDNA 208 bp	55.1
Short down	CACGTTAGCTCCGACACGGAT		56.5
Short up	GCGAAGGCGCTTTTCTAATTTA	16S rDNA 121 bp	54.9
Chtinok	GTTGAGACCATCCACATCAAGTATG		56.8
MT for	CACCATTAGCACCCAAAGCT	mtDNA 1023 bp	51.9
MT rev	CTGTTAAAAGTGCATACCGCCA		54.6
16S1 F	CCCGCCTGTTTACCAAAAACAT	mtDNA 250 bp	56.8
16S1 R	AAGCTCCATAGGGTCTTCTCGTC		54.7

**Table 2 microorganisms-09-00935-t002:** PCR programs.

Primers	Program
PNEU S/PNEU Nok PNEU S/Short down Chp up/Chp down	94 °C—3 min, 35 × (94 °C—45 s, 50 °C—1 min, 72 °C—1 min), 72 °C—5 min, 14 °C
Short up/Chtinok	94 °C—3 min, 30 × (94 °C—45 s, 50 °C—1 min, 72 °C—1 min), 72 °C—5 min, 14 °C
MT for/MT rev	94 °C—5 min, 30 × (94 °C—1 min, 54 °C—1 min, 72 °C—1 min), 72 °C—3 min, 14 °C
16S1 F/16S1 R	94 °C—5 min, 30 × (94 °C—1 min, 55 °C—1 min, 72 °C—1 min), 72 °C—3 min, 14 °C

**Table 3 microorganisms-09-00935-t003:** Pros and cons of NPCR assays used for the identification of *C. pneumoniae*.

Reference	Source	Primers/Tm (°C)	PCRProductSize (bp)	Comments	Gene
Fukano, 2004 ^b^ [28]Al-Aydie et al., 2016 ^b^ [29]	EB, B,NPS	53K-1 ATGATCGCGGTTTCTGTTGCCA (61.1 °C) 53K-2 GAGCGACGTTTTGTTGCATCTC (56.3 °C) 53K-3 TGTCCAAGCGGTGAAACAAG (53.5 °C)53K-4 CAACCGTGACCCATTTACTG (50.3 °C)	499	Primers specific to *C. pneumoniae,* but only 80% DNA sequence identity to the counterparts from other *Chlamydia* infecting humans.	CopB

239
LaBiche et al., 2001 ^b^ [30]Nadareishvili et al., 2001 ^b^ [31]Reszka et al., 2008 ^b^ [32]Assar et al., 2016 ^b^ [33]	AP, AW	TTATTCACCGTC**C**TACAGCAGAAA (55.6 °C)GGGGGTTCAGG**G**ATCATTTGT (52.7 °C)TTACGAAACGGCATTACAACGGCTAGAAATCAAT (67.4 °C)TATGGCATATCCGCTTCGGGAACGAT (65.7 °C)	404	Primers specific to *C. pneumoniae*, but only 80% DNA sequence identity to the counterparts from other *Chlamydia* infecting humans, mismatch in primer sequence.	rpoB

214
Maass et al., 1997 ^b^ [34]Yazouli et al., 2018 ^a^ [35]	PBMC,AP	HL-1 GTTGTTCATGAAGGCCTACT (45.8 °C)HR-1 TGCATAACCTA**C**GGTGTGTT (42.4 °C)In-1 AGTTGAGCATATTCGTGAGG (46.8 °C) In-2 TTTATTTCCGTGTCGTCCAG (50.2 °C)	437	Primers specific to *C. pneumoniae*, but only 80% DNA sequence identity to the counterparts from other *Chlamydia* infecting humans, wrong primer sequence insertion of C.	rpoB


128
Messmer et al., 1997 ^b^ [36]	B, LT,F, CS,NPS	1S ACGGA**AT**AAT**GAC**TTCG**G** (44.5 °C)1A TACC**T**GGTACGCTCAATT (47.9 °C)2S ATAAT**GAC**TTCG**GTTG**TT**A**TT (42.9 °C)2A CGTCATCGCCTTGGTGGGCTT (62.8 °C)	436	Primers specific to some *Chlamydia*; low Tm; extreme Tm difference between the second pair of primers; polymorphism in primer binding site.	16S rRNA


221
Wilson et al., 1996 ^b^ [37]	C	OTF CGATCGCTAATACCGAATGT**A**G**TG ** (54.8 °C)OTR **T**TAGCCAATCTCT**CTTAT**TC**C**CAG (53.0 °C)INF A**AA**GC**C**CACCAAG**GCG**ATG (57.2 °C)INR AAAGTGCTTTACAACCCTA**A** (45.5 °C)	317	Primers specific to *C. pneumoniae*, but not to other *Chlamydia* human pathogens; extreme Tm difference between the second pair of primers; polymorphism in primer binding site.	16S rRNA


178
Black et al., 1994 ^b^ [38]Blasi et al., 1996 ^b^ [39]	AP, TS	1 **AT**AAT**GAC**TTCG**GTTG**TT**A**T (40.6 °C)2 **TATAAATA**GGTTGAGTCAAC (35.9 °C)3 **A**G**TG**T**AA**TT**A**GGCATC**TA**A**TA**T (39.9 °C)4 **G**C**TGT**ATTTCT**A**C**A**G**TT**G (33.6 °C)	1397	Primers specific to *C. pneumoniae*, but not to other *Chlamydia* human pathogens; low Tm; large PCR product; polymorphism in primer binding site.	16S rRNA


858
Tong and Sillis, 1992 ^b^ [40]Sessa et al., 2001 ^b^ [41]Apfalter et al., 2002 ^b^ [42] Kumar et al., 2016 ^b^ [43]	SP	CP1 T**T**A**C**AAGCCTTGCCT**G**T**A**GG (49.9 °C)CP2 GCGATCCCAAAT**G**TTTA**A**GGC (55.2 °C)CPC TTATTAAT**T**GATGG**T**AC**A**AT**A** (37.6 °C) CPD **ATCT**A**CGG**C**A**GTAGTAT**AG**T**T** (40 °C)	333	Primers specific to *C. pneumoniae*, but not to other *Chlamydia* human pathogens; low Tm; large PCR product; polymorphism in primer binding site.	MOMP*ompA*


NA	207
Nystrom-Rosander et al., 1997 ^a^ [44]	AW	Cpn ATGAC**AA**C**T**G**T**AGAAAT**ACA**G**C ** (42.0 °C)Cpn B CGCCTCTCTCC**TATAAAT** (40.7 °C)Cpn 1 CCGCAAGGACA**T**ATACACAGG (54.5 °C)Cpn 2 CCAGTTCGGATTGTAGTCTGC (51.4 °C)	463	Second pair of primers at the same strand; low Tm; polymorphism in primer binding site.	16S rRNA


289
Kaltenbock et al., 1997 ^a^ [45]Sachse and Hotzel, 2003 ^b^ [46]	V	191 GCIYTITGGGARTGYGGITGYGCIAC (66.5 °C)371 TTAGAAICKGAATTGIGCRTTIAYGTGIGCIGC (65.2 °C)201 GGIGCWGMITTCCAATAYGCICARTC (59.0 °C)336 CCRCAAGMTTTTCTRGAYTTCAWYTTGTTR (59.2 °C)	582	Primers specific to *Chlamydia* human pathogens; despite degeneration, polymorphism in primer binding site.	MOMP*ompA*


443
Lindholt et al., 1998 ^b^ [47]	AW	OutA.TACTGGATCCGCTGCTGCAAACTATACTAC (61.8 °C)OutB CTGTTGCTACGCCAGCGTCTGTTG (62.7 °C)InA GTAGATAGACCTAACCCGGCCTACAATAAG (59.3 °C)InB TAGTACCTTTAACTCCGAATAAACCAACGA (59.0 °C)	496	Primers specific to *C. pneumoniae*, but not to other *Chlamydia* human pathogens; polymorphism in primer binding site.	MOMPompa



189
Contini et al., 2018 ^b^ [48]Mahony et al., 2000 ^b^ [49]	CVTPBMC	CpnA TGACAACTGTAGAAATACAGC (42.0 °C)CpnB CGCCTCTCTCCTATAAAT (40.7 °C)TW50 AGTCCCGCAACGAGCGCA (59.9 °C)TW51 GCTGACACGCCATTACTA (43.7 °C)	463	CpnA and CpnB primers unspecific to Chlamydia in silico; specific to *C. pneumoniae*, but not to other *Chlamydia* human pathogens; extreme Tm difference.	16S rRNA


217
Lienard et al., 2011 ^a^ [50]	NPS	panCh16F2 CCGCCAACACTGGGACT (51.1 °C)panCh16R2 GGAGTTAGCCGGTGCTTCTTTA (51.9 °C)panCh16S FAMCTACGGGAGGCTCAGTCGAGAATC-BHQ	208	Primers designed in conserved regions; proved to be nonspecific by sequencing; 16S RNA gene was amplified from *Schaalia odontolytica*, *Actinomyces sp.*	16S rRNA

Degenerate nucleotides: K = G, T; M = A, C; R = A, G; W = A, T; Y = C, T; I = Inosine; ^a^ confirmed by sequencing; ^b^ not sequenced; **CGG**—mismatch/polymorphism is bold underlined. Abbreviations: AP—atherosclerotic plaques; AW—aortic wall; B—blood; EB—elementary bodies; NPS—nasopharyngeal specimen; PBMC—peripheral blood mononuclear cells; F—feces; LT—lung tissues; CS—cloacal swabs; C—culture; TS—throat swabs; SP—sputum; SV—sclerotic valves; V—varia; NA—nasopharyngeal aspirate; CVT—chorionic villi tissues

**Table 4 microorganisms-09-00935-t004:** *Chlamydia pneumoniae* detection limits.

Method/Source	Dilution of DSM Culture	Number of Cell Equivalents in the Sample/µL	Cell Equivalent in PCR Reaction
LNPCR/cells	10^−4^	0.5	0.25
LNPCR/sputum ^a^	10^−2^ *	100	0.034
SNPCR/sputum ^a^	10^−3^	10	0.0034
RTPCR/cells	10^−1^	0.5	2.5
RTPCR/sputum ^a^	10^0–1 b^	5 × 10^3^	3.4

LNPCR—long-nested PCR for 461 bp amplicon; SNPCR—short-nested PCR for 121 bp amplicon; RTPCR—real-time PCR about 120 bp amplicon; ***** the same result after repeated thawing and freezing. ^a^ Alterations between dilution of DSM culture and cell equivalent are due to the differences in the DNA isolation procedure. ^b^ Ct 35 in 10^−1^ was above the internal standard (Ct 31). Each sample was spiked, extracted, and tested by PCR separately in two independent experiments, always with the same result.

## Data Availability

Written informed consent has been obtained from the patient(s) to publish this paper” if applicable.

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
