# Peer review of "Reliable and Sensitive Nested PCR for the Detection of Chlamydia in Sputum"

_microorganisms, 2021, doi:10.3390/microorganisms9050935_

Round 1

Reviewer 1 Report

The manuscript entitled “Reliable and sensitive nested PCR for the detection of Chlamydia in sputum” by Dr. molejová describes a newly nested PCR method, for the detection of Chlamydia DNA in sputum samples.

The manuscript is in general well written and organized, I do not have any particular comment, only few observations:

The authors confirmed NPCR data by sequencing on positive samples. Have the authors molecularly analyzed the sequences in order to determine the presence of putative polymorphisms among samples? If not, this analysis should be performed.

Line 46-56 The Nested‐PCR method has also previously been employed for the detection of genomic DNA sequences of Chlamydia trachomatis in chorionic Villi samples (PMID: 30078192). In particular, 6S generis, 16s‐chtracho primer sets, targeting the 16S gene region of Chlamydia trachomatis has been employed. This information along with the reference should be included as background.

Line 128 rate [49, Fig. S1). --> please correct this typo error

Author Response

The authors confirmed NPCR data by sequencing on positive samples. Have the authors molecularly analyzed the sequences in order to determine the presence of putative polymorphisms among samples? If not, this analysis should be performed.

Yes, we have done it. Figure 2 shows the phylogenetic relationships of species of the order Chlamydiales and our specimens and depicts polymorphism in our samples concerening LNPCR. Polymorphism in SNPCR depicts Figure S5 Alignment of SNPCR sequences. Polymorphic sites are grey shadowed.

Line 46-56 The Nested‐PCR method has also previously been employed for the detection of genomic DNA sequences of Chlamydia trachomatis in chorionic Villi samples (PMID: 30078192). In particular, 6S generis, 16s‐chtracho primer sets, targeting the 16S gene region of Chlamydia trachomatis has been employed. This information along with the reference should be included as background.

The reference PMID: 30078192 is “Contini, C.; Rotondo, J.C.; Magagnoli, F.; Maritati, M.; Seraceni, S.; Graziano, A.; Poggi, A.; Capucci, R.; Vesce, F.; Tognon, M.; et al. Investigation on silent bacterial infections in specimens from pregnant women afected by spontaneous miscarriage. J. Cell. Physiol. 2018, 234, 100–107”. However, they used 16S primers designed in “Mahony, J. B., S. Chong, B. K. Coombes, M. Smieja, and A. Petrich. 2000. Analytical sensitivity, reproducibility of results, and clinical performance of five PCR assays for detecting Chlamydia pneumoniae DNA in peripheral blood mononuclear cells. J. Clin. Microbiol. 38:2622–2627”. These primers were evaluated from the point of view of specificity and efficiency of the PCR in silico  and implemented to the Table 3.

Reviewer 2 Report

The manuscript details and compares various primer sets previously used to identify various Chlamydia species in clinical samples. The overall broad spectrum amplification abilities and specificity of primer sets were compared.  Upon comparing the 16s rDNA sequence the authors design new primers to amplify a region of the 16s rDNA sequence from multiple Chlamydia strains.  This was done successfully.

Overall the manuscript is well written and the science and methods are sound. The results are important and can impact diagnostics.  

Only minor revisions are requested. Please see comments below:

Figure 1, can the authors clarify what samples are in the different lanes in the agarose gel of PCR products? This is unclear.

There is a lack of consistency when italicizing pathogen name. Some are noted below in this review but there are many additional ones that need fixing.

Specific comments:

Line 61: Where is HPL s.r.o. Microbiological Laboratory located?  Add this info in.  Also, what does HPL stand for?

Lines 63, 64: Where are the Faculty of Natural Sciences and the Faculty of Physical Education and Sports located? Add this information in.

Line 91: Chlamydia pneumoniae  should be in italics

Line 92: ompA should be in italics

Line 116&168: in silico should be in italics

Lines 117, 119/120: C. pneumoniae should be in italics

Lines 126/127: ompa should be ompA and in italics

Line 134: Schaalia odontolytica should be in italics

Line 138: C. trachomatis should be in italics

Lines 160-162: italicize all bacterial names

Lines 171-172: can the authors clarify as to where the primers that were designed would bind the 16s rDNA gene?  A schematic or figure would be very helpful.  The supplementary info has details and sequence info but a small supplementary figure showing locations would be helpful and can easily be incorporated into Figure 1

Lines 178/179/188/216/260: C. pneumoniae should be in italics

Line 179: The authors state that "TWAR-183 culture" was used. Can you clarify, do you mean the DNA?  Otherwise if you are adding culture then this would be colony PCR?

Author Response

Figure 1, can the authors clarify what samples are in the different lanes in the agarose gel of PCR products? This is unclear.

This part we fixed.

Figure 1. Threshold value of the NPCR assay for C. pneumoniae detection in sputa. Lines: M λ/PstI size marker; 10-4 – 101 500 µl of negative sputum was spiked with 1 µl of 10-4 – 101 dilution of C. pneumoniae TWAR-183 original culture; DNA was isolated as described in the Materials and Methods section, and 0.5 µl was added to the first PCR reaction, amplified by external primers, and 1 µl was amplified in the second reaction; NC negative control. A. PNEU S/PNEU Nok; Chp up/Chp down primers. Size of PCR product is 461 bp. B. PNEU S/Short down; Short up/a Chtinok primers. Size of PCR product is 121 bp.

There is a lack of consistency when italicizing pathogen name. Some are noted below in this review but there are many additional ones that need fixing.

We apologize for this drawback. It happened after copy paste from word to Microsoft Word template. Hopefully we fixed them all.

Specific comments:

Line 61: Where is HPL s.r.o. Microbiological Laboratory located?  Add this info in.  Also, what does HPL stand for?

Lines 63, 64: Where are the Faculty of Natural Sciences and the Faculty of Physical Education and Sports located? Add this information in.

This part we fixed.

Sputum samples from 10 patients with pneumonia (mostly atypical) were obtained from the HPL s.r.o. Microbiological Laboratory (Bratislava, Slovakia) for routine plating tests (four males and six females; mean age: 42; range, 24 to 68 yr.). Another 156 samples were provided by volunteers from the Faculty of Natural Sciences and the Faculty of Physical Education and Sports (Comenius university, Bratislava, Slovakia).